# Multivariate description of gait changes in a mouse model of peripheral nerve injury and trauma

Bilal A. Naved[1,2], Shuling Han[2], Kyle M. Koss[2,3], Mary J. Kando[4], Jiao-Jing Wang [2], Craig Weiss [4], Maya G. Passman [5], Jason A. Wertheim[3], Yuan Luo[6]*, Zheng J. Zhang [2]*

1 Department of Biomedical Engineering, McCormick School of Engineering, Northwestern University, Evanston, IL, United States of America, 2 Comprehensive Transplant Center, Department of Surgery, Northwestern University Feinberg School of Medicine, Chicago, IL, United States of America, 3 Department of Surgery, College of Medicine, University of Arizona, Tucson, Arizona, United States of America, 4 Behavioral Phenotyping Core, Northwestern University Feinberg School of Medicine, Chicago, IL, United States of America, 5 Barnard College, Columbia University, New York, NY United States of America, 6 Division of Health and Biomedical Informatics, Department of Preventive Medicine, Northwestern University Feinberg School of Medicine, Chicago, IL, United States of America

* yuan.luo@northwestern.edu (YL); zjzhang@northwestern.edu (ZJZ)

**Data Availability Statement:** The datasets generated during and/or analyzed during the current study are publicly available at a data

## Abstract

### Objective

Animal models of nerve injury are important for studying nerve injury and repair, particularly for interventions that cannot be studied in humans. However, the vast majority of gait analysis in animals has been limited to univariate analysis even though gait data is highly multidimensional. As a result, little is known about how various spatiotemporal components of the gait relate to each other in the context of peripheral nerve injury and trauma. We hypothesize that a multivariate characterization of gait will reveal relationships among spatiotemporal components of gait with biological relevance to peripheral nerve injury and trauma. We further hypothesize that legitimate relationships among said components will allow for more accurate classification among distinct gait phenotypes than if attempted with univariate analysis alone.

### Methods

DigiGait data was collected of mice across groups representing increasing degrees of damage to the neuromusculoskeletal sequence of gait; that is (a) healthy controls, (b) nerve damage only via total nerve transection + reconnection of the femoral and sciatic nerves, and (c) nerve, muscle, and bone damage via total hind-limb transplantation. Multivariate relationships among the 30+ spatiotemporal measures were evaluated using exploratory factor analysis and forward feature selection to identify the features and latent factors that best described gait phenotypes. The identified features were then used to train classifier models and compared to a model trained with features identified using only univariate analysis.

repository and can be found at the following DOI
(10.6084/m9.figshare.25546822).

**Funding:** Research reported in this publication was
supported by the National Institutes of Health
under Award Numbers F30DK123985, and
T32GM008152 (BAN); CTC Transplant Innovation
Endowment grant (110-5442000) (Zhang), DOD
Department of the Army: W81XWH2110862
(Zhang), McCormick Foundation/Northwestern
Memorial Hospital (Zhang and Wertheim), Julius
N. Frankel Foundation via Northwestern Memorial
Foundation (Zhang, Han, and Wang); American
Heart Association: 20POST35210774 and the
Canadian Institute for Health Research: RN409371
- 430628 (Koss); and R01LM013337 (Luo). The
content is solely the responsibility of the authors
and does not necessarily represent the official
views of the National Institutes of Health. Funding
sources had no roles in study design, collection,
analysis, and interpretation of the data, in the
writing of the report, nor in the decision to submit
the article for publication. There was no additional
external funding received for this study.

**Competing interests:** The authors have declared
that no competing interests exist.

**Abbreviations:** VCA, vascularized composite
allotransplantation; PCA, principal component
analysis; SFI, sciatic function index; ALS,
amyotrophic lateral sclerosis; IACUC, institutional
animal care and use committee; MCE,
misclassification error; CDF, cumulative
distribution function; PQDA, pseudo-quadratic
discriminant analysis.

## Results

10–15 features relevant to describing gait in the context of increasing degrees of traumatic
peripheral nerve injury were identified. Factor analysis uncovered relationships among the
identified features and enabled the extrapolation of a set of latent factors that further
described the distinct gait phenotypes. The latent factors tied to biological differences
among the groups (e.g. alterations to the anatomical configuration of the limb due to trans-
plantation or aberrant fine motor function due to peripheral nerve injury). Models trained
using the identified features generated values that could be used to distinguish among
pathophysiological states with high statistical significance ($p < .001$) and accuracy (>80%)
as compared to univariate analysis alone.

## Conclusion

This is the first performance evaluation of a multivariate approach to gait analysis and the
first demonstration of superior performance as compared to univariate gait analysis in ani-
mals. It is also the first study to use multivariate statistics to characterize and distinguish
among different gradations of gait deficit in animals. This study contributes a comprehen-
sive, multivariate characterization pipeline for application in the study of any pathologies in
which gait is a quantitative translational outcome metric.

## Introduction

In the study of human gait, methods of statistical evaluation include multivariate processes of
feature extraction and selection [1]. Animal models are valuable for studying conditions and
treatments that are difficult to study and experiment with in humans (e.g. traumatic nerve
injuries, limb replacement, limb transplantation, tissue regeneration strategies) [2].

More than 20 million people in the United States are estimated to have some form of
peripheral nerve damage [3]. However, available treatments for peripheral neuropathies are
limited to correcting underlying causes, promoting healthy lifestyle habits, controlling
immune responses + inflammation with medication, engaging in physical therapy, using reha-
bilitative devices, receiving surgery, and/or adopting behavioral strategies to cope [3]. There
are no available treatments that specifically aim to promote nerve repair and regeneration.
Rodent models are versatile for assessing novel approaches to promote nerve repair ranging
from tissue engineering constructs to cutting-edge surgical interventions.

There are widely adopted methods for evaluating success when studying nerve injury and
repair. These include methods for assessing cellular appearance (histopathologic examination)
[4], nerve conduction (nerve conduction velocity) [5], nerve-muscle connection (electromyog-
raphy) [6], sensory function (Hargreaves) [7], and motor function (Rotarod, DigitGait, Cat-
Walk, swimming, climbing) [8, 9].

Ultimately, the lower extremity's prime function is to enable locomotion. In human gait
analysis, studies have investigated multivariate techniques to describe relationships among
measurable features of gait [1]. These relationships help statistically characterize gait pheno-
types and are used to select features for measurement, classification, and prediction [1]. Imple-
menting a similar level of statistical rigor to animal gait studies represents a novel problem
requiring further refinement and investigation.

In rodents, the study of locomotion via treadmill gait (DigiGait™) or free ambulating (CatWalk™) systems provides high levels of spatial and temporal detail, measuring over 30+ individual spatial or temporal gait parameters [10–14]. Analyses of data measured by these systems are largely limited to traditional, univariate, multiple-hypothesis testing or feature selection [15–18]. Usually this involves comparing the means of individual output parameters for statistical significance and applying some correction for multiple-hypothesis testing [19, 20]. Such univariate study enables conclusions that do not account for possible relationships among individual gait parameters. A machine learning model can also be trained using feature selection techniques that omit the examination of multivariate relationships between features [21]. Thus, discoveries in rodent models, to date, have been primarily limited to a portion of the interplay that makes up the animal's overall gait.

These limited, univariate gait studies span a variety of rodent models of gait deficit including those of metabolic (diabetic neuropathy) [22], degenerative (ALS) [23], traumatic (nerve transection / limb transplantation) [24], congenital [25], and other etiologies. While these diseases may result in dysfunctional gait, it is unlikely that the spatiotemporal details of their dysfunction are the same. For locomotion, indices such as the Sciatic Function Index (SFI) have been developed via multivariate linear regression models to compare healthy and dysfunctional gait [26]. However, the SFI is a dimensionless number and does not reveal additional insight into exact relationships between specific spatial and temporal measures [27]. In short, what does it really mean if the SFI index is statistically different between two gait states? How do we describe more than just a "difference" between two states?

A comprehensive description of the rodent's gait would require a methodology that can factor relationships among all 30+ measures and prioritize those that contribute to gait phenomena rather than noise. We hypothesize that out of all spatiotemporal measures of rodent gait, there exists a discrete subset sufficient to uniquely describe the gait of three distinct physiological states: (1) healthy animals, (2) those with nerve injury due to a complete nerve transection and re-attachment, and (3) those with nerve, muscle, and bone injury due to total limb transplantation. Furthermore, we hypothesize that multivariate characterization will reveal relationships that encode and describe the different gait phenotypes in a biologically consistent manner unlike univariate analysis. Finally, we hypothesize that the revealed relationships will enable the training of a classifier model that can discriminate between gait states with higher accuracy than models trained with features identified via univariate analysis.

## Methods

### Animals and experimental groups

43 male B6 mice, 8 to 10 weeks old (20g), were obtained from The Jackson Laboratory (Bar Harbor, ME, USA) and were group-housed. Male mice were used due to their larger size, thereby having easier vascular access.

There were three groups: 17 animals in the negative control group received no treatment. 12 animals in the experimental group received only nerve damage (neurorrhaphy) i.e. a complete nerve transection of the femoral and sciatic nerves with re-coaptation. 14 animals in the second experimental group received a total hind-limb transplant. Animals receiving a procedure received only one procedure on their left hind limb.

The motivation for assessing these three groups was to establish a gradient of increasing neuromuscular damage. In doing so, the goal was to characterize how spatio-temporal gait parameters were altered accordingly. Thereby, investigating how musculoskeletal damage, in addition to nerve injury, alters the statistical characterization compared to just the contribution of nerve injury alone. In other words, groups were chosen to represent increasing degrees of damage to the neuromusculoskeletal sequence of healthy gait.

Additionally, while clinical experience to date suggests it is unlikely that clinical lower limb transplantation will become common, the hind-limb transplantation model was still used in this study for four main reasons: (1) as part of an increasing gradient of damage on which to study the utility of multivariate characterization to identifying biologically relevant features and relationships, (2) to investigate the value of multivariate characterization as a novel and proposed new standard for gait analysis in a therapeutic intervention and model that is difficult, albeit, impossible to study in humans, (3) to establish and characterize this model and statistical pipeline as one of potential benefit to research in functional nerve recovery for upper extremity transplantation, and (4) to maximize the degree of damage and thereby spatiotemporal changes to the limb in the interest of evaluating this new method of gait characterization at its limits.

All experimental procedures were conducted according to an IACUC-approved protocol (protocol IS00003195) and were conducted in accordance with the National Research Council's Guide for the Care and Use of Laboratory Animals. All surgery was performed with animals kept on a self-regulating heating pad and under isofluorane anesthesia. The depth of anesthesia was examined before surgery with a toe pinch response and subsequent checks at 15-minute intervals during the surgery. Post-operatively, the animals were kept in their cages, with a portion of the cage on a warm pad. The animals were checked on every hour for the first 3 post-op hours and then 4 times a day for two days for any signs of surgical complications. To minimize suffering, pain was managed via local anesthetics (bupivacaine or lidocaine) at the surgical site with a multi-modal analgesic regimen of Buprenorphine-ER and meloxicam to last an additional 3 days. At the conclusion of all experimental timepoints animals were sacrificed via Isofluorane euthanasia with confirmatory bilateral thoracotomy.

## Mouse hind-limb transplant

The murine hind-limb transplantation was performed using the surgical technique modified from the one previously described by our lab [24] and Furtmuller et al [28]. The donor procedure took ~45 minutes per procedure and the recipient procedure took ~2 hours per procedure. The isogenic transplants required no immunosuppressive regimen. Briefly, for donor right hind limb harvest, after adequate anesthesia, the right hind limb was shaved and prepped with 70% ethanol plus povidone iodine, followed by a circumferential incision in the mid-thigh. The femoral neurovascular bundle was carefully dissected. The nerve, artery, and vein were separated and were then transected distally at the level of the superficial epigastric artery. The ventral and dorsal muscle groups were divided proximally to the mid-thigh, thereby separating the remainder of the hind limb from the animal's body. The hind-limb graft was perfused with heparinized Ringer's solution and stored in a Ringer's solution-containing dish at 4˚C. For the recipient limb implantation, the recipient native right hind limb amputation was performed similarly to the donor right hind limb harvest. The donor limb graft was anatomically aligned and the recipient's femur bone was connected with the donor femur bone using an intramedullary stainless-steel rod. Muscle coaptation was accomplished using 6–0 Polysorb suture material. Vessel anastomoses were established using the cuff technique. Continuity of the sciatic and femoral nerves was established with 11–0 sutures. Then, the skin was closed using 6–0 monofilament sutures (Ethilon). After the surgery, the animal was then regularly monitored for at least 4 hours before returning to the housing facility.

## Mouse model of sciatic and femoral nerve transection injury

The sciatic and femoral nerve transection model took ~30 min. to complete. The mice were anesthetized and prepped as described in the previous section. An incision of approximately 8

mm in length in the right posterolateral hind limb was made. The intermuscular septum was gently dissected. The sciatic nerve was exposed and separated from its sheath before being cut using sharp micro scissors. The two nerve stumps were carefully repositioned, and the epineurium was connected using an 11–0 suture. No retraction or torsion was observed, and the stumps fit well without any gaps between them. Similar procedures were performed on the femoral nerve. Subsequently, the skin was suture closed as described in the previous section. The animal was placed under a heating lamp and allowed to recover in its cage. Regular monitoring was conducted for at least 4 hours before the animal was returned to the housing facility.

## Digigait procedure

After the respective procedures required for each experimental group, the animals were allowed to recover and reacclimate for two weeks before beginning data collection on the treadmill system. The mice were kept in an animal holding room and acclimated in the Behavioral Phenotyping Core before experiments. Gait was measured using the DigiGait™ Imaging System (Mouse Specifics Inc.) at three different speeds (10, 17, and 24 cm/s). Footprints from a 3–4 second video clip were analyzed for each speed using Digigait™ Analysis version 15. We used the software to quantify 30+ gait indices.

## Statistical analysis

**Dimensionality reduction.** All analysis was conducted in MATLAB. Exploratory analysis to reduce and identify the primary dimensions responsible for describing rodent gait was conducted via feature selection and factor analysis techniques. Feature selection was conducted using two contrasting approaches: (a) a traditional univariate analysis with Bonferroni correction and (b) a multivariate, forward sequential feature selection with cross-validation [29].

**Feature selection.** In this study we implement a simple filter by applying univariate criteria separately on each feature with Bonferroni correction. This was done as a baseline to replicate typical (i.e. non-multivariate) analysis of gait performed across the field. This baseline is important for evaluating the hypothesis that multivariate characterization will identify feature sets and feature relationships that better describe gait states and will lead to more accurate classification. We also apply a multivariate, forward sequential feature selection in a wrapper fashion to find important features with the goal of minimizing misclassification error (MCE) of our learning algorithm. This was done to explore the hypothesis that multivariate characterization will reveal relationships that encode and describe the different gait phenotypes in a biologically consistent manner unlike univariate analysis; and that the revealed relationships will enable the training of a classifier model that can discriminate between gait states with higher accuracy than models trained on features identified via univariate analysis. Embedded selection was then applied during model training to further confirm the ideal feature set. During the feature selection procedure we applied 10-fold cross-validation to the training set. We ensured that the same animal was not included in both the training and test sets of each individual fold. While feature selection identified the measured parameters to use in the model it did not describe how the selected parameters may relate to each other.

**Factor analysis.** To learn more about the exact relationships among features, factor analysis was conducted. Understanding the exact relationships among features is important to exploring the hypotheses that said relationships encode and describe the different gait phenotypes in a biologically consistent manner; and that the revealed relationships will enable the training of a classifier model that can discriminate between gait states with higher accuracy than models trained with features identified via univariate analysis. In the factor analysis

model measured variables are dependent upon a smaller number of unobserved or latent factors [30]. The coefficients on these latent factors are called "loadings". An added component to account for noise called "specific variance" is included. Average specific variance and factor loading as a function of number of latent factors was plotted to determine the number of factors that would be reasonable to assume for analysis. A loading cutoff greater than 0.6 or less than -0.6 was then used to identify the important features within each factor grouping.

**Machine learning pipeline: Cross validation, model training, and accuracy calculation.** *Definitions within study context.* Machine learning: computational statistical algorithms that learn from data labeled as coming from healthy, nerve transected, or limb transplanted mice and using those learnings to understand the features and relationships that encode patterns specific to those respective groups to enable the training of models that can be used for various purposes like classification.

Training of classifier models: computational statistical model development via machine learning methods using data labeled from healthy, nerve transected, or limb transplanted mice. The classifier models are encoded by features and relationships identified by the machine learning methods. Once trained, new, unlabeled data from animals of uncertain gait quality can be input into the model. The model will then return numerical values representing the likelihood that the input data is characteristic of a healthy, nerve-transected, or limb-transplanted animal and its gait. The output values can be used to classify animals with known identities into respective groups. They can also be used to quantify the quality of the statistical characterization conducted on each gait phenotype.

*Cross validation.* Training and testing sets were defined by randomly picking 80% as training and 20% as testing data. 10 randomly selected training and testing sets were defined to perform 10-fold cross validation. We ensured that the same animal was not included in both the training and test sets within a respective fold.

*Model training and performance evaluation.* Various classifier model architectures were evaluated using the features identified from the feature selection and dimensional reduction techniques described above. Based on the feature selection results 16 features were chosen to be included in model training. The evaluated architectures included discriminant analyses, random forest, and support vector machine. Regardless of the model type, training was performed using the results of the multivariate feature selection process described above. Outputs of all models were either 0 = dysfunctional gait (surgery) or 1 = healthy gait (control). Average model accuracy, precision, recall, and F-score were measured at the conclusion of the cross-validated model training process. Models trained via multivariate feature selection were compared for their performance with a model trained via univariate feature selection to explore the hypothesis that a classifier model trained on features identified via multivariate analysis can discriminate between gait states with higher accuracy than models trained with features identified via univariate analysis.

## Results

### Surgical results

A total of 30 surgeries were performed in this study with a high success rate (87%). The experimental group receiving neurorraphy contained 14 animals, the group receiving total hindlimb transplant contained 12 animals, and the control group contained 17 animals.

Attention to the vascular anastomoses, neural repair, and bone coaptation was crucial to maximizing the viability of the limb transplant model. The isografted mice lived for the course of the study and none of them showed signs of rejection. The peripheral nerve transection model involved femoral and sciatic nerve transection and coaptation only. All mice in this

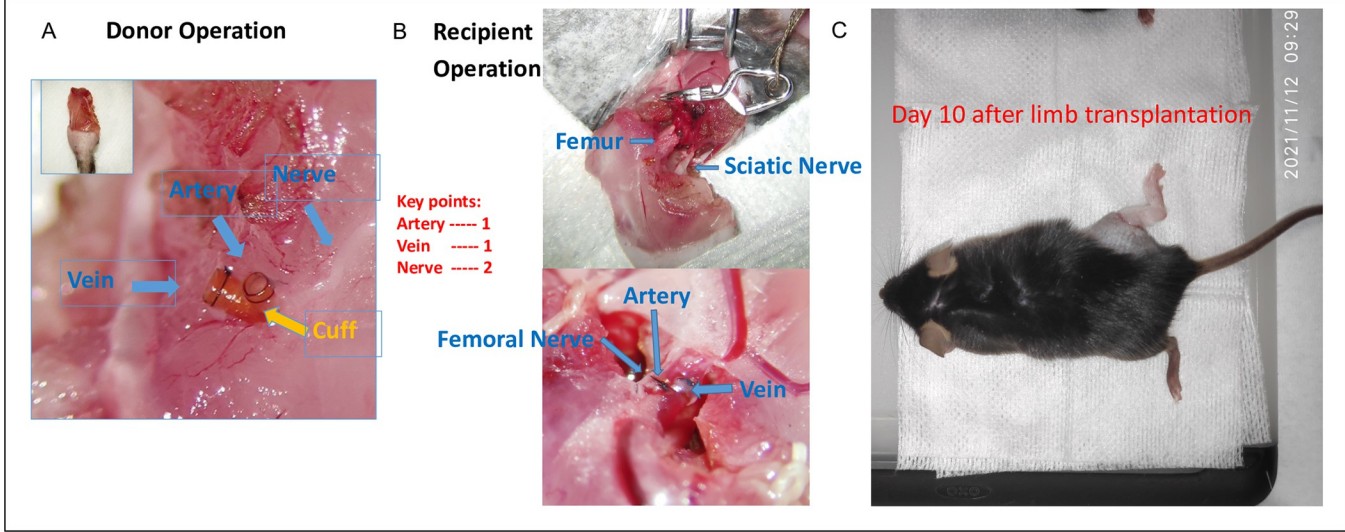

**Fig 1. Hind-limb transplant and peripheral nerve transection model design. (A)** The donor's femoral neurovascular bundle was carefully dissected. The femoral nerve, artery, and vein were carefully separated and transected distally at the level of the superficial epigastric artery. **(B)** The donor limb graft was anatomically aligned such that the recipient's femur bone was connected with the donor femur bone using an intramedullary stainless-steel rod, vessel anastomoses were done with the cuff shown in panel A, and the sciatic and femoral nerves were connected with 11–0 sutures. **(C)** Panel C shows a mouse with a completed hind-limb transplantation. After the procedure the mice were allowed to recover and acclimate for two weeks before proceeding with DigiGait evaluation. The peripheral nerve transection model involved total transection of the femoral and sciatic nerves in the same way it was done in the hind-limb transplantation model.

group also lived for the course of the study. Both models used a nerve cuff to hold the coapted nerves in place. Operative design and representative results are shown in **Fig 1**.

## Feature selection

**Univariate analysis.** Traditional gait studies conduct univariate analysis, which was done here as a basis for comparison. We first plotted the empirical cumulative distribution function (CDF) of the *p*-values. The CDF of the *p*-values visualizes the percentage of feature means, when compared for statistical difference via t-test that lie under the critical value. Applying Bonferroni correction for multiple hypothesis testing determined a critical *p*-value of .001. In comparing pathological gait from limb transplantation to healthy controls, 28% of features had statistical significance (**Fig 2**), whicih amounts to 9 parameters (**Table 1**). In comparing pathological gait from nerve transection injury alone, 44% of features had statistical significance (**Fig 2**), which amounts to 14 parameters (**Table 1**). Comparing the means of every feature 1:1 with the control and then comparing the set of statistically significant features between the two degrees of gait deficit revealed a high degree of shared features with statistical significance (**Table 1**). However, via univariate analysis we learn little about how the identified features relate to each other to describe the respective gait states.

**Univariate filter feature selection.** Measuring and plotting misclassification error (MCE) as a function of an increasing number included features is a univariate approach to determining the ideal number of features to include in model training. To illustrate, we compute MCE for a discriminant analysis model between 2 to 32 features and plot MCE accordingly (**Fig 3**). Note, in this simple, univariate feature addition approach, using 30 features minimizes MCE. Ultimately, the smallest MCE achieved with this method was 0.17 indicating a potential model accuracy of 83% in discriminating healthy gait from pathological gait due to limb transplantation. However, this is a model trained on one holdout set with likely overfitting from using 30

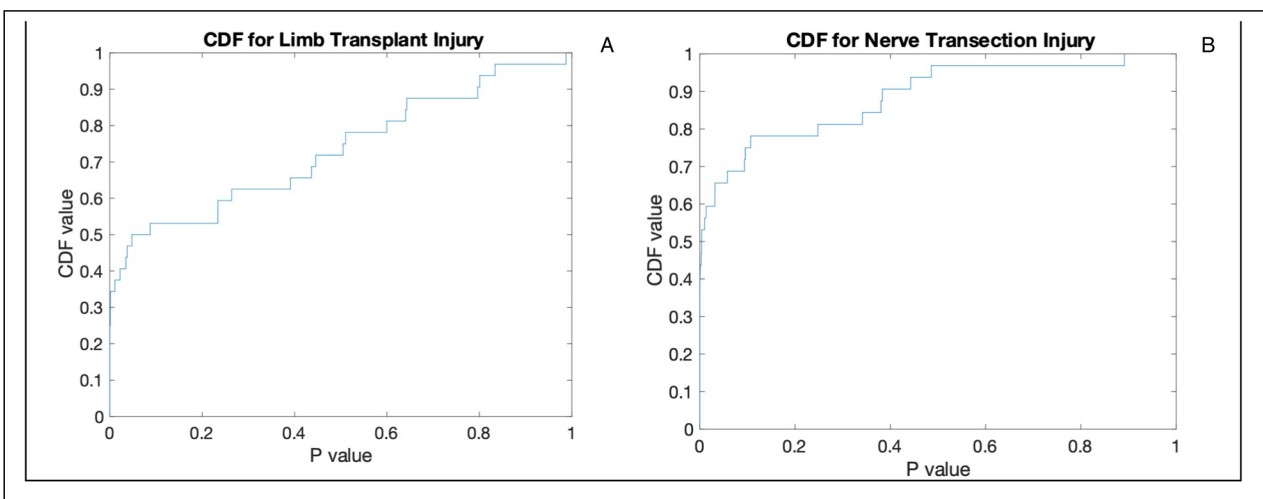

**Fig 2.** Cumulative distribution function showing (**A**) the percent of features that are under a significant p-value when comparing their means between healthy versus transplanted animals with t-tests. Boneferroni correction determines a cutoff of 0.05 / 32 = 0.001. 28% of features have p-values below the .001 threshold and (**B**) the percent of features that are under a significant p-value when comparing their means between healthy versus nerve transection injury animals with t-tests. Boneferroni correction determines a cutoff of 0.05 / 32 = 0.001. 44% of features have p-values below the .001 threshold. The features below threshold in (a) and (b) are identified in columns 1 and 2 of Table 1 are establish a baseline of features selected univariately for comparison to those identified via multivariate selection. The value in multivariate feature selection will be explored quantitatively in subsequent figures.

of the 32 available features. It is likely that this is modeling noise and it is unclear how much generalizable gait phenomena is being modeled. Thus, multivariate (wrapper) methods for feature selection were explored.

**Multivariate wrapper feature selection with cross validation.** The univariate approach to filter feature selection described above omits relationships between features and is representative of how most rodent gait studies have evaluated their data. Moreover, choosing features purely based on their ranking in filter selection can result in the modeling of redundant information. For example, features 2 and 8 have a linear correlation coefficient of 1.0. This is because these two features, are by definition, related (%SwingStride = 1 - %StanceStride).

Sequential feature selection is a multivariate approach to selecting a subset of features that works by sequentially adding features in various relationships to the model and monitoring exactly which feature relationships lead to decreases in MCE until it is minimized. When classifying between healthy gait and pathological gait due to limb transplantation, our results show that 10 to 12 features are optimal for minimizing MCE (smallest MCE achieved = 0.12 (**Fig 3**). For classifying between healthy and pathological gait due to nerve transection injury, our results show that 16 features achieve the minimum possible cross-validation MCE of 0.10. This is a 5–7% improvement in MCE achieved using ~20 fewer features than the simple filter approach above. Note how MCE increases beyond the critical numbers of features indicating overfitting and supporting the following hypotheses: (a) there is a critical subset of features that more accurately describes healthy versus pathological gait and (b) the subset identified via multivariate characterization more generally describes gait when compared to the subset identified via univariate characterization as evidenced by superior model performance and less overfitting. The selected features are shown in **Table 1.**

However, when examining the selected parameters, it is unclear how they may relate to each other. Despite the compelling classification results from adding multivariate feature selection, the process does not reveal exactly how these features relate to each. A technique like factor analysis is needed to identify precisely which features relate to each other.

**Table 1. Statistically significant features according to univariate analysis done with Boneferroni correction vs. from forward selection (multivariate).**

| Healthy vs. nerve transection (univariate) | Healthy vs. limb transplant (univariate) | Features from forward selection of healthy vs. nerve transection (multivariate) | Features from forward selection of healthy vs. transplant (multivariate) |
|---|---|---|---|
| **1. Swing** | **Swing** | 1. Brake | **1. %SwingStride** |
| **2. %SwingStride** | **%SwingStride** | **2. %BrakeStride** | **2. %PropelStride** |
| 3. %BrakeStride | **%PropelStride** | **3. Stance/Swing** | **3. %StanceStride** |
| **4. %StanceStride** | **%StanceStride** | **4. StrideLength** | **4. StrideLength** |
| 5. Stride | **Stance** | 5. Stride Frequency | **5. Absolute PawAngle** |
| **6. %BrakeStance** | | **6. Paw Angle Variability** | **6. Paw Angle Variability** |
| 7. %PropelStance | | 7. #Steps | **7. Stride Length CV** |
| **8. Stance/Swing** | **Stance/Swing** | **8. Stride Length CV** | 8. Paw Area at Peak Stance in sq. cm |
| 9. Stride Frequency | | **9. Paw Area Variability at Peak Stance in sq. cm** | **9. Paw Area Variability at Peak Stance in sq. cm** |
| **10. Paw Angle Variability** | **Paw Angle Variability** | **10. Gait Symmetry** | **10. Gait Symmetry** |
| 11. #Steps | | 11. MIN dA/dT | 11. MAX dA/dT |
| 12. MIN dA/dT | | **12. Overlap Distance** | **12. Overlap Distance** |
| **13. Overlap Distance** | **Overlap Distance** | **13. PawPlacementPositioning[PPP]** | **13. PawPlacementPositioning[PPP]** |
| **14. PawPlacementPositioning [PPP]** | **PawPlacementPositioning [PPP]** | 14. Midline Distance | 14. Ataxia Coefficient |
| | | **15. Axis Distance** | **15. Axis Distance** |
| | | 16. Belt Speed | |

Based on the univariate analysis, we see three types of features in both pathological states: those related to phase (absolute and relative measures), measures of the paw, and measures of symmetry. Features selected from forward selection (multivariate analysis) in both pathological states revealing features related to stride length and how it varies, measures of the paw, measures of symmetry, and measures of phase. Features are listed in rank order based on their contribution to characterizing each pathology as selected by the forward selection algorithm. Bolded features are ones that are shared between phenotypes. Notice that univariate and multivariate analysis have significant differences in the relevant features with certain areas of overlap. Within each type of analysis (univariate vs. multivariate), the two pathological states share numerous features. The features identified via univariate and multivariate characterization respectively will be used to build classifier models to quantify our ability to classify between gait states in the subsequent figures and tables.

## Factor analysis

A prior gait study proposed a multivariate rodent gait model to represent groupings of variables that related to each other and better describe gait. Lambert et al. proposed that the multidimensional DigiGait data in rats with olivocerebellar ataxia can be reduced to three uncorrelated groupings of variables, or common factors, termed rhythmicity, thrust, and contact [31]. Similarly, factor analysis allows us to understand how the directly measured features (i.e. the collected data) compose potential latent (or unmeasurable) factors that more directly describe the phenomena of interest. To determine the number of likely factors, the average specific variation and loading were plotted as a function of factor number (**Fig 4**).

In **Fig 4**, the red line represents the average specific variation, and the blue line represents the average loading. The results confirm that at 6 latent factors, the loading does not increase, while the specific variation continues to decrease, which indicates overfitting. Maximal loading is viewed using 6 factors, while reaching an inflection point in minimizing specific variation. Thus, 6 factors were used to continue the comparative analysis between the latent factors that characterize pathological gait due to transplantation versus nerve transection injury (**Table 2**). **Fig 4**'s bottom-right two panels show the calculated feature loadings of greater than 0.6 or less than -0.6, which was identified as a significant feature in the respective factor grouping. Note that Stance is the only feature shared between two factors. Otherwise, the factors are each composed of unique features.

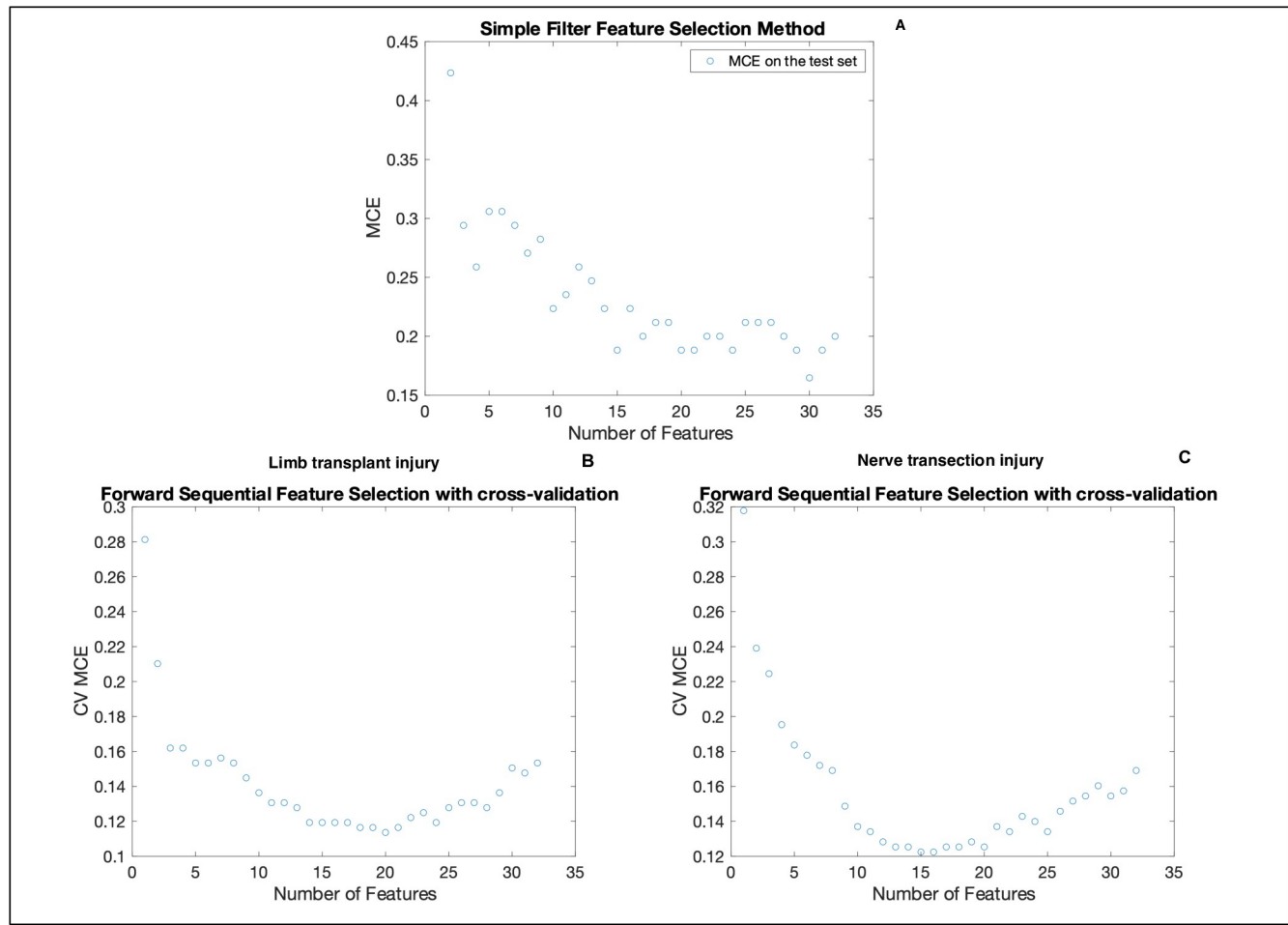

**Fig 3. Contrasting univariate and multivariate feature selection methods for their ability to minimize misclassification error (MCE) as a function of number of features included in the model. Top: (A)** Misclassification error (MCE) as a function of number of features calculated using a pseudo-quadratic discriminant analysis, holdout set, and a simple filter method that does not account for any interactions. The lowest achievable MCE was around 0.16 when including 30 features. **Bottom:** Multivariate, forward feature selection with cross-validation to minimize misclassification error when classifying healthy gait from pathological gait was conducted to find the optimal combination of features that minimizes MCE. This approach takes interactions between features into account and uses a pseudoquadratic discriminant analysis. The lowest MCE was **(B)** ~0.12 using 10–14 features when classifying **limb transplantation** injury and **(C)** ~0.12 using 10–14 features when classifying just **nerve transection injury.** Notice how the MCE has a much more consistent and predictable pattern as features are added in comparison to the univariate, simple filter approach referenced in **(A)**.

Table 2 shows the latent factors and the features they comprise for pathological gait due to limb transplantation. Table 2 also shows the latent factors and the features they are composed of for pathological gait due to nerve transection injury. The bolded latent factors in each sub-table (Factor 4 and Factor 6 respectively) are composed of different features between the two groups. The other 5 latent factors are composed of the same (or highly similar) features. The distinction is that pathological gait due to limb transplantation is distinguished by the relationship between the absolute paw angle and midline distance. Pathological gait due to nerve transection injury is distinguished by the relationship between paw area at peak stance in sq. cm, paw area variability at peak stance in sq. cm, and MIN dA/dT. The high degree of overlap of feature groupings between the two pathologies suggests that much of the statistical description of gait in these two sources of traumatic nerve injury is the same with a significant point of difference being factors 4 and 6 respectively (i.e. the two bolded columns) (Table 2).

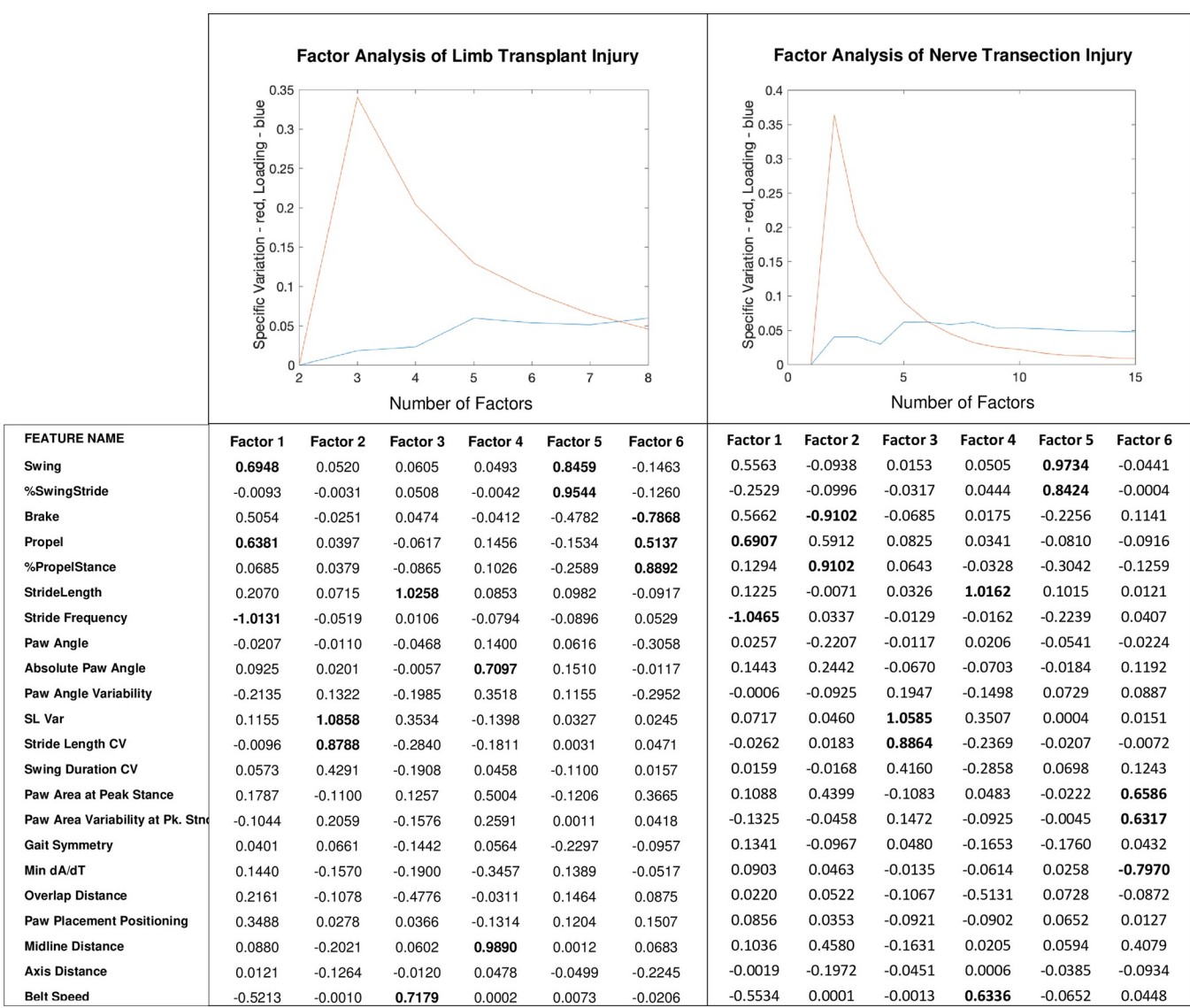

| FEATURE NAME | Factor 1 | Factor 2 | Factor 3 | Factor 4 | Factor 5 | Factor 6 | Factor 1 | Factor 2 | Factor 3 | Factor 4 | Factor 5 | Factor 6 |
|---|---|---|---|---|---|---|---|---|---|---|---|---|
| Swing | **0.6948** | 0.0520 | 0.0605 | 0.0493 | **0.8459** | -0.1463 | 0.5563 | -0.0938 | 0.0153 | 0.0505 | **0.9734** | -0.0441 |
| %SwingStride | -0.0093 | -0.0031 | 0.0508 | -0.0042 | **0.9544** | -0.1260 | -0.2529 | -0.0996 | -0.0317 | 0.0444 | **0.8424** | -0.0004 |
| Brake | 0.5054 | -0.0251 | 0.0474 | -0.0412 | -0.4782 | **-0.7868** | 0.5662 | **-0.9102** | -0.0685 | 0.0175 | -0.2256 | 0.1141 |
| Propel | **0.6381** | 0.0397 | -0.0617 | 0.1456 | -0.1534 | 0.5137 | **0.6907** | 0.5912 | 0.0825 | 0.0341 | -0.0810 | -0.0916 |
| %PropelStance | 0.0685 | 0.0379 | -0.0865 | 0.1026 | -0.2589 | 0.8892 | 0.1294 | **0.9102** | 0.0643 | -0.0328 | -0.3042 | -0.1259 |
| StrideLength | 0.2070 | 0.0715 | **1.0258** | 0.0853 | 0.0982 | -0.0917 | 0.1225 | -0.0071 | 0.0326 | **1.0162** | 0.1015 | 0.0121 |
| Stride Frequency | **-1.0131** | -0.0519 | 0.0106 | -0.0794 | -0.0896 | 0.0529 | **-1.0465** | 0.0337 | -0.0129 | -0.0162 | -0.2239 | 0.0407 |
| Paw Angle | -0.0207 | -0.0110 | -0.0468 | 0.1400 | 0.0616 | -0.3058 | 0.0257 | -0.2207 | -0.0117 | 0.0206 | -0.0541 | -0.0224 |
| Absolute Paw Angle | 0.0925 | 0.0201 | -0.0057 | **0.7097** | 0.1510 | -0.0117 | 0.1443 | 0.2442 | -0.0670 | -0.0703 | -0.0184 | 0.1192 |
| Paw Angle Variability | -0.2135 | 0.1322 | -0.1985 | 0.3518 | 0.1155 | -0.2952 | -0.0006 | -0.0925 | 0.1947 | -0.1498 | 0.0729 | 0.0887 |
| SL Var | 0.1155 | **1.0858** | 0.3534 | -0.1398 | 0.0327 | 0.0245 | 0.0717 | 0.0460 | **1.0585** | 0.3507 | 0.0004 | 0.0151 |
| Stride Length CV | -0.0096 | **0.8788** | -0.2840 | -0.1811 | 0.0031 | 0.0471 | -0.0262 | 0.0183 | **0.8864** | -0.2369 | -0.0207 | -0.0072 |
| Swing Duration CV | 0.0573 | 0.4291 | -0.1908 | 0.0458 | -0.1100 | 0.0157 | 0.0159 | -0.0168 | 0.4160 | -0.2858 | 0.0698 | 0.1243 |
| Paw Area at Peak Stance | 0.1787 | -0.1100 | 0.1257 | 0.5004 | -0.1206 | 0.3665 | 0.1088 | 0.4399 | -0.1083 | 0.0483 | -0.0222 | **0.6586** |
| Paw Area Variability at Pk. Stnc | -0.1044 | 0.2059 | -0.1576 | 0.2591 | 0.0011 | 0.0418 | -0.1325 | -0.0458 | 0.1472 | -0.0925 | -0.0045 | **0.6317** |
| Gait Symmetry | 0.0401 | 0.0661 | -0.1442 | 0.0564 | -0.2297 | -0.0957 | 0.1341 | -0.0967 | 0.0480 | -0.1653 | -0.1760 | 0.0432 |
| Min dA/dT | 0.1440 | -0.1570 | -0.1900 | -0.3457 | 0.1389 | -0.0517 | 0.0903 | 0.0463 | -0.0135 | -0.0614 | 0.0258 | **-0.7970** |
| Overlap Distance | 0.2161 | -0.1078 | -0.4776 | -0.0311 | 0.1464 | 0.0875 | 0.0220 | 0.0522 | -0.1067 | -0.5131 | 0.0728 | -0.0872 |
| Paw Placement Positioning | 0.3488 | 0.0278 | 0.0366 | -0.1314 | 0.1204 | 0.1507 | 0.0856 | 0.0353 | -0.0921 | -0.0902 | 0.0652 | 0.0127 |
| Midline Distance | 0.0880 | -0.2021 | 0.0602 | **0.9890** | 0.0012 | 0.0683 | 0.1036 | 0.4580 | -0.1631 | 0.0205 | 0.0594 | 0.4079 |
| Axis Distance | 0.0121 | -0.1264 | -0.0120 | 0.0478 | -0.0499 | -0.2245 | -0.0019 | -0.1972 | -0.0451 | 0.0006 | -0.0385 | -0.0934 |
| Belt Speed | -0.5213 | -0.0010 | **0.7179** | 0.0002 | 0.0073 | -0.0206 | -0.5534 | 0.0001 | -0.0013 | **0.6336** | -0.0652 | 0.0448 |

**Fig 4. Scree plot and average loading as a function of number of factors on healthy vs. limb transplanted gait data.** The red line represents average specific variation and the blue line represents average loading (top two panels). At around 6 factors is when average loading plateaus and when average specific variation is at its inflection point. Thus, we examined the first 6 factors for their constituent features (bottom three panels). Using a cutoff of +/- 0.6, loadings were examined to identify feature groupings that composed the 6 most relevant factors. Features with loadings of greater than 0.6 or less than -0.6 were marked in bold in the bottom panels.

It can be suggested that animals with limb transplantation have more variation in the anatomy of their transplanted paw in relation to the midline of their bodies. In contrast, animals that received nerve transection injury and reattachment do not demonstrate this same variation. This is consistent with the underlying biology: animals with a total limb transplantation likely have an alteration in the anatomical configuration of their paw related to their midline while animals receiving only nerve transection injury might not. Additionally, the data suggests that animals with exclusively nerve transection injury have a discernible variation in how the contact of their paw relates to their ability to make fine motor movements (i.e. jerks or dA/dT) while animals receiving a total limb transplant are likely to have this masked by gross motor defects related to the extensive musculoskeletal injury that accompanies the nerve injury from the transplant.

**Table 2. Factor analysis results of limb transplant data in contrast to only nerve transection injury data showing the features that make up each factor.**

| Factor Analysis Results of Data From Pathological Gait Due To Limb Transplantation | | | | | |
|---|---|---|---|---|---|
| **Factor 1** | **Factor 2** | **Factor 3** | **Factor 4** | **Factor 5** | **Factor 6** |
| Swing | SL Var | StrideLength | **Absolute Paw Angle** | Swing | Brake |
| Propel | Stride Length CV | Belt Speed | **Midline Distance** | %SwingStride | Propel |
| Stride Frequency | | | | | %PropelStance |
| Factor Analysis Results of Data From Pathological Gait Due To Nerve Transection | | | | | |
| **Factor 1** | **Factor 2** | **Factor 3** | **Factor 4** | **Factor 5** | **Factor 6** |
| Propel | Brake | SL Var | Stride Length | Swing | **Paw Area at Peak Stance in sq. cm** |
| Stride Frequency | %Propel Stance | Stride Length | Belt Speed | %SwingStride | **Paw Area Variability at Peak Stance in sq. cm** |
| | | | | | **Min dA/dT** |
| Latent factors that describe gait in the setting of traumatic nerve injury | | | | | |
| **Relationship of stride length and speed** | **Stride length and its variation** | | **Measures of the paw** | **Measures of symmetry** | **Phases of the stride** |

Notice key similarities (unbolded) and differences (**bolded**) between the two datasets. The bottom two rows describe the latent or implied factors that make sense of the various groups of observable measures shown above. These latent factors are an interpretation of the factor groupings shown above. The factor groupings were identified via factor analysis, which quantifies the relationship between individual, measurable features and can be used to identify the most highly grouped features

## Training a machine learning classifier to discriminate between gait phenotypes

Using the features identified via multivariate feature selection and confirmed by factor analysis we assessed four different model architectures (**Table 3**). Accuracy in distinguishing between healthy gait and pathological gait due to any form of peripheral nerve injury (**Table 3A**) ranged from 0.75–0.91, precision from 0.78–0.93, recall from 0.77–0.91, and F-Score from 0.77–0.92. The ensemble model (i.e. boosted classification trees), had the highest-performing metrics in distinguishing between healthy gait and pathological gait due to any form of peripheral nerve injury.

**Table 3. Evaluating the performance of 4 different model architectures in distinguishing between healthy and pathological phenotypes of gait.**

| A. Performance in Distinguishing Peripheral Gait Deficit from Healthy Gait | Accuracy | Precision | Recall | F-Score |
|---|---|---|---|---|
| **Random Forest** | 0.7294 | 0.7560 | 0.7521 | 0.7492 |
| **Discriminant Analysis** | 0.7477 | 0.8022 | 0.7634 | 0.7742 |
| **Support Vector Machine** | 0.7744 | 0.8108 | 0.7915 | 0.7948 |
| **Regression** | 0.7868 | 0.8570 | 0.7826 | 0.8130 |
| **Ensemble** | 0.9099 | 0.9283 | 0.9086 | 0.9165 |
| **B. Performance in Distinguishing Between Two Phenotypes of Peripheral Gait Deficit: Limb Transplant from only Nerve Transection** | Accuracy | Precision | Recall | F-Score |
| **Random Forest** | 0.6435 | 0.7072 | 0.6852 | 0.6878 |
| **Discriminant Analysis** | 0.7165 | 0.7827 | 0.7188 | 0.7388 |
| **Support Vector Machine** | 0.6987 | 0.7806 | 0.7341 | 0.7457 |
| **Regression** | 0.7237 | 0.7882 | 0.7270 | 0.7456 |
| **Ensemble** | 0.8780 | 0.9263 | 0.8781 | 0.8984 |

Using the identified features from feature selection + factor analysis 4 different model architectures were evaluated for their accuracy, precision, recall, and F-score in their ability to distinguish (**A**) healthy gait from gait deficit due to peripheral nerve injury and (**B**) gait deficit due limb transplantation from gait deficit due to total nerve transection alone.

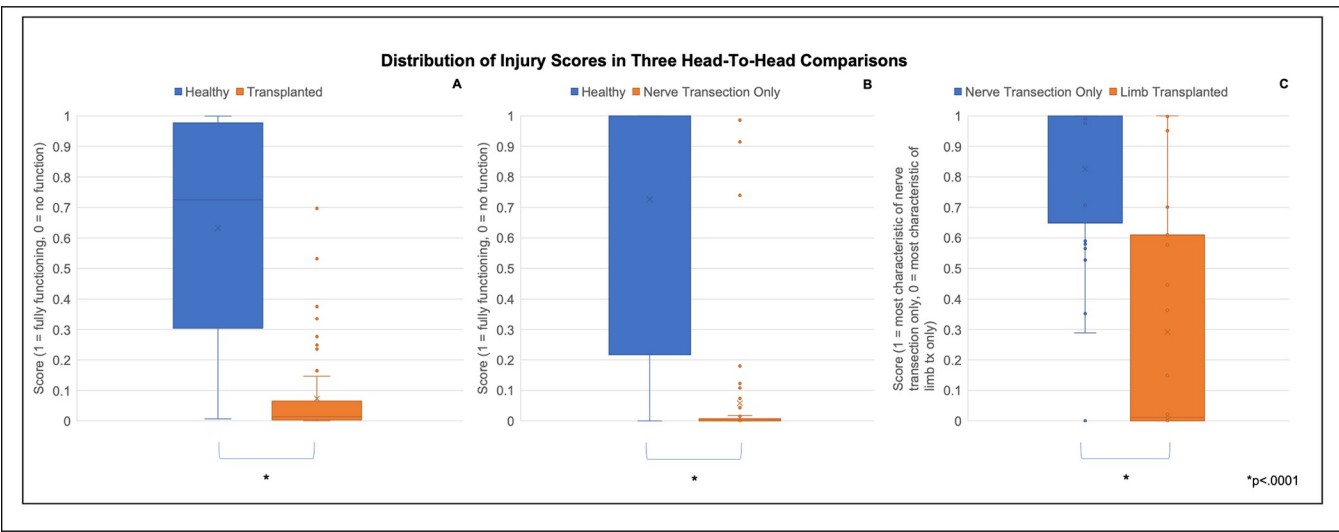

**Fig 5. Box and whisker plot showing output values from a discriminant analysis model.** The discriminant analysis model was used to generate values for animals with various phenotypes and compare (A) healthy gait to dysfunctional gait due to limb transplantation (B), healthy gait to dysfunctional gait due to nerve transection alone, and (C) dysfunctional gait due to nerve transection from dysfunctional gait due to limb transplantation. Independent, two-tailed t-tests were conducted with correction for multiple hypothesis testing to reveal high statistical significance among groups (p < 0.0001).

Accuracy in distinguishing between types of peripheral nerve injury (i.e., hind-limb transplant vs. total nerve transection) ranged from 0.64–0.88, precision from 0.71–0.93, recall from 0.69–0.88, and F-score from 0.69–0.90. The ensemble model again was the highest-performing architecture in distinguishing between different types of peripheral nerve injury.

Values outputted by the model on the holdout set were graphed as a box and whisker plot. A strong statistical difference in output distribution between healthy versus injured animals as well as between pathological states was observed (p < .0001) (**Fig 5**). Sample confusion matrices from one iteration were calculated and included as well (**S1 and S2 Figs in S1 File**).

## Discussion

The goal of this study was to use multivariate statistics to learn about the features, relationships, and factors that are most descriptive of gait in different pathological states. In doing so, we explored the hypothesis that a multivariate characterization of gait will reveal novel relationships between spatiotemporal components specifically in the context of peripheral nerve injury and trauma. Gait data from DigiGait's system includes 30+ different spatiotemporal features or outputs. It is important to understand which features are most descriptive of specific pathological states. We called these features primary dimensions and defined them as those features most relevant to describing relationships within the gait data. We hypothesized that these relationships were deterministic of specific phenotypes, which contrasts with features that are not related to describing specific phenotypes and may conversely model noise.

Modeling noise can result in overfitting that detracts from statistical learning of the patterns that describe the phenotype of interest. Feature selection algorithms enable one to test various feature subsets from the original feature set and learn which subset is most descriptive of the underlying phenomena. Feature selection algorithms can be roughly grouped into three categories: filter, wrapper, and embedded methods [29]. Univariate filter methods select the feature subsets via multiple hypothesis testing without involving a learning algorithm [32]. Multivariate wrapper methods iteratively evaluate a variety of relationships between features

and use the performance of a learning algorithm to evaluate each candidate feature subset [33]. Embedded methods determine feature importance as part of a model training process [34].

In **Fig 3** the wrapper-based, multivariate feature selection resulted in the usage of 10 to 15 select features in the model and thereby decreased the MCE of the model by more than 20%. This result supports the hypothesis that accounting for relationships between components of the gait would allow for more accurate classification between distinct gait phenotypes than univariate studies alone (**Table 2**).

This accounting for multivariate relationships is largely absent from the field of animal gait analysis. Examining the features selected in a multivariate manner in **Table 2** suggests that the two pathologies differ in their relationship to jerk (dA/dT). Nerve transection injury was better defined by minimum jerk (e.g. fine motor movements) while limb transplant injury was better defined by maximum jerk (e.g. gross motor movements). Clinically, the ability to identify characteristic measures of different degrees of injury could be relevant to evaluation, monitoring, and rehabilitation. It may inform the development of sensors capable of more precisely monitoring parameters like jerk. It may also help suggest quantitatively discernible differences in underlying (patho)physiology for further evaluation and focus.

Further exploring the potential importance of the single feature of jerk in distinguishing between these two gait phenotypes reveals that minimum vs. maximum jerk may be of particular importance (**Table 2**). While the two gradations of injury share many features as shown in Table 2, they are distinguished by the presence of minimum vs. maximum jerk. This difference suggests a hypothesis to investigate further: that nerve transection injury's single source of variation to jerk is the nerve injury itself, resulting in a more discernible impact on fine motor movements (i.e. minimum dA/dT), whereas limb transplant injury, being a product of nerve + muscle + bone injury, contributes to a wider set variation and deficiencies in jerk, resulting in a more discernible impact to gross motor movements (i.e. maximum dA/dT). Translating this idea to a clinical setting, the neurologic physical exam includes components for evaluating fine motor and gross motor components of the gait [35]. However, the typical exam is at best semi-quantitative and does not measure jerk in a manner that would lend itself to statistical evaluation like the assessments performed in this study [35]. There exist quantitative methods for measuring changes in locomotive jerk in humans [36]. Perhaps there is value to doing so when evaluating the degree of peripheral nerve injury in settings of new presenting trauma or in monitoring the recovery of gait over time in response to trauma.

The ranking of the features in **Table 2** may provide additional insight. It is listed in rank order of features selected for their proportion of contribution to characterizing each pathology. It is interesting to see the high degree of overlap in ranking between the nerve transection and limb transplant. Additionally, it is interesting to see how the features have some natural groupings. Measures of phase are primary, then secondarily measures of the paw, and tertiarily measures of symmetry. Also tertiary are measures of fine and gross motor movements respectively. While the measures of phase and the paw could be mutually characteristic of healthy gait and/ or both etiologies of traumatic injury, we see that the tertiary measures are where significant difference between the two gait phenotypes may be resolved. The level of exploratory statistical description in the prior few paragraphs is largely absent from rodent studies across the field, including some of the latest that employ gait analysis [18, 25, 37]. Thus, we hypothesize that multivariate statistics has generalizable potential in reducing observational bias to current preclinical gait research protocols in the scientific community.

Towards this hypothesis, a recent study contributes evidence toward using gait as a quantitative translational outcome metric for therapeutic development in Angelman syndrome and other genetic neurodevelopmental syndromes [25]. This study contributes a comprehensive, multivariate characterization pipeline for application in the study of any pathologies in which

gait is a quantitative translational outcome metric. Another study was comparable in its level of statistical description [31], coincidentally doing so in a genetic neurodevelopmental model of cerebellar ataxia [31]. Lambert et al. supported the value of multivariate characterization of gait by conducting feature extraction on highly dimensional gait data from a central nerve injury model of cerebellar ataxia. The authors hypothesized certain indirect, common factors that characterize gait and underlie the more directly measurable features. These common factors included thrust, rhythmicity, and contact area and were directly useful in discriminating between animals with a central lesion vs. those without [31]. In this study, we observed potential patterns in the selected features and hypothesized that pathological gait due to traumatic nerve injury could also be characterized by certain latent factors that are determined by and composed of the selected features.

The identified latent factors fell into similar categories as the ones identified by sequential feature selection. These were phasic parameters that represent either absolute or relative measures of the phases of each stride, symmetric parameters representing some measure of the gait symmetry, and regional parameters, which represent some measure of features specific to certain anatomical regions of the limb (i.e. the paw). The common factors and their feature compositions suggest that rodent gait is statistically characterized by a careful coordination between sub-components of the limb (e.g. specific anatomical parameters), the limb itself (e.g. phasic parameters), and the torso/remainder of the animal (e.g. symmetric parameters).

When contrasting the factors that are statistically meaningful to describing both types of traumatic nerve injury, we learn that the relationship between stride length and speed is important to both injury models (**Table 2**). Moreover, variation in the length of the stride is also important. Measures of the paw are also important factors in both models. However, the specific measures of the paw are consistent with biological differences in the respective pathologies. For example, limb transplantation injury was better characterized by the relationship of the paw to the midline, which may be due to loss of mechanical integrity and thereby positioning of the limb in relation to the area surrounding the transplant itself. In contrast, the nerve transection injury model was better characterized by fine motor movements of the paw (i.e. MIN dA/dT). This suggests that finer measures are more discernible when there is only nerve injury as compared to nerve + muscle + bone injury (**Table 2**). The results from factor analysis were consistent with the results from sequential feature selection and helped validate that the two gradations of peripheral nerve injury shared numerous important features. The results also displayed key differences related to fine vs. gross motor movements in each model respectively. Factor analysis allowed for a deeper understanding of the exact relationships among features in contrast to a simple rank list of features as provided by many feature selection techniques. This level of statistical description is currently lacking from gait studies in animal models. The many insights revealed support the hypothesis that multivariate gait analysis provides a more comprehensive statistical description of individual gait states than univariate analysis. In the clinic, employing such factor analysis techniques could similarly provide a window into better describing nuances of different patients' gaits. This study offers a methodology that may be applied to existing DigiGait datasets to conduct similar exploratory multivariate analysis to reduce observational bias from traditional univariate methods. Evaluating the generalizability of this methodology on other etiologies of gait deficit is the subject of a subsequent study.

Lastly, we explored the hypothesis that characteristic feature relationships of distinct gait phenotypes could be used to train a classifier capable of distinguishing between said phenotypes. The trained classifier distinguishing individual animals for whether their gait was more likely healthy or pathological and discriminated between the two pathologies with high accuracy (**Table 3**). The model outputted a distribution of values with highly significant difference

in their means (p < .0001) (**Fig 5**). The model was built by encoding the features and relationships uncovered to be most important in describing the gait pathologies respectively. Via the outputted values, we statistically distinguished between the two different traumatic pathologies with high statistical significance (p<0.0001). This is the first system to multivariately describe and distinguish between gradations of limb injury in a murine gait model. As a new methodology within rodent gait research protocols, one limitation of this study is its limited scope. On one hand, the multivariate statistics demonstrated in this study offer a convenient way for researchers to conceptualize meaningful groupings of features. On the other hand, clear biological tie-ins and strong statistical significance in classifier performance when applied to gait research protocols across the scientific community stands to be evaluated. We have identified datasets of DigiGait data available at our Behavioral Phenotyping Core from investigators studying central causes of gait disorder (e.g. stroke, transient ischemic attack) for us to apply the pipeline to and are pursuing said inquiry. Academic researchers can do the same by accessing the code available at our GitHub repository and citing this manuscript should any code be used towards future publications (https://github.com/luoyuanlab/gait). Industry professionals must reach out to the corresponding author to make use of the code.

## Conclusion

There is no precedent for using multivariate statistics to characterize and distinguish between different etiologies of gait deficit. According to a PubMed search, hundreds of gait studies in animals have been published over the past two decades. The majority of them being univariate studies. This investigation paves the way for future studies of gait pathology across a variety of etiologies to conduct similar characterization. The statistical pipeline developed in this study represents an application of state-of-the-art methods to multivariately describe and quantify functional gait outcomes in a biologically consistent manner to a gradation of increasing injury. Development of this method may allow for more detailed characterization of functional outcomes in response to various therapeutic strategies of repairing peripheral nerve injury. There is an opportunity for method developers to contribute to time series characterization and classification. However, a barrier to time series study is the lack of an animal model of gait deficit that is confirmed to heal and recover a statistically significant amount of gait function to use as a positive control. More broadly, there is also an opportunity for methods developers to apply the pipeline developed here in the study of other causes of gait deficit (e.g. central, metabolic, vascular, degenerative, (auto)immune, genetic) and evaluate this method's generalizability. The means to do so being offered in this study.

## Supporting information

**S1 File. Confusion Matrices of Varying Classifier Architectures (S1 Fig. Peripheral Injury vs. Control and S2 Fig. Nerve Transection vs. Limb Transplant).** All confusion matrices are from a single iteration of ten randomly selected training-testing splits. Performance metrics reported in the manuscript are the average of those ten.
(DOCX)

## Acknowledgments

The mouse hind-limb transplantation and the peripheral nerve injury surgical procedures were performed by the "Microsurgery & Preclinical Research Core" at Northwestern University Comprehensive Transplant Center. Gait analysis was done in the Behavioral Phenotyping Core at Northwestern University.

## Author Contributions

**Conceptualization:** Bilal A. Naved, Shuling Han, Kyle M. Koss, Mary J. Kando, Jiao-Jing Wang, Craig Weiss, Jason A. Wertheim, Yuan Luo, Zheng J. Zhang.

**Data curation:** Bilal A. Naved, Mary J. Kando, Maya G. Passman.

**Formal analysis:** Bilal A. Naved, Maya G. Passman.

**Funding acquisition:** Jason A. Wertheim, Zheng J. Zhang.

**Investigation:** Bilal A. Naved, Shuling Han, Kyle M. Koss, Mary J. Kando, Jiao-Jing Wang, Craig Weiss, Jason A. Wertheim, Yuan Luo, Zheng J. Zhang.

**Methodology:** Bilal A. Naved, Shuling Han, Mary J. Kando, Craig Weiss, Yuan Luo, Zheng J. Zhang.

**Project administration:** Bilal A. Naved, Kyle M. Koss, Craig Weiss, Jason A. Wertheim, Yuan Luo, Zheng J. Zhang.

**Resources:** Craig Weiss, Jason A. Wertheim, Zheng J. Zhang.

**Software:** Bilal A. Naved.

**Supervision:** Jason A. Wertheim, Yuan Luo, Zheng J. Zhang.

**Validation:** Bilal A. Naved, Kyle M. Koss, Jason A. Wertheim, Yuan Luo, Zheng J. Zhang.

**Visualization:** Bilal A. Naved.

**Writing – original draft:** Bilal A. Naved, Shuling Han, Mary J. Kando, Maya G. Passman, Jason A. Wertheim, Yuan Luo, Zheng J. Zhang.

**Writing – review & editing:** Bilal A. Naved, Shuling Han, Mary J. Kando, Craig Weiss, Maya G. Passman, Jason A. Wertheim, Yuan Luo, Zheng J. Zhang.

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
