## [Decision Letter · Decision Letter 0]

1 Apr 2024

PONE-D-23-36233Multivariate description and scoring of neuromotor changes in a mouse model of peripheral nerve injuryPLOS ONE

Dear Dr. Zhang,

Thank you for submitting your manuscript to PLOS ONE. After careful consideration, we feel that it has merit but does not fully meet PLOS ONE’s publication criteria as it currently stands. Therefore, we invite you to submit a revised version of the manuscript that addresses the points raised during the review process.

We look forward to receiving your revised manuscript.

Kind regards,

Aliaa Rehan Youssef, PhD

Academic Editor

PLOS ONE

Journal Requirements:

2. To comply with PLOS ONE submissions requirements, in your Methods section, please provide additional information regarding the experiments involving animals and ensure you have included details on (1) methods of sacrifice,  and (2) efforts to alleviate suffering.

"Research reported in this publication was supported by the National Institutes of Health under Award Numbers F30DK123985, and T32GM008152 (BAN); DOD Department of the Army: W81XWH2110862 (Zhang), McCormick Foundation/Northwestern Memorial Hospital (Zhang and Werthiem), Julius N. Frankel Foundation via Northwestern Memorial Foundation (Zhang, Han, and Wang); American Heart Association and CIHR (Koss); and R01LM013337 (Luo). The content is solely the responsibility of the authors and does not necessarily represent the official views of the National Institutes of Health. Funding sources had no roles in study design, collection, analysis, and interpretation of the data, in the writing of the report, nor in the decision to submit the article for publication."

Reviewers' comments:

Reviewer's Responses to Questions

**Comments to the Author**

1. Is the manuscript technically sound, and do the data support the conclusions?

Reviewer #1: No

Reviewer #2: Yes

Reviewer #3: Partly

2. Has the statistical analysis been performed appropriately and rigorously? 

Reviewer #1: Yes

Reviewer #2: N/A

Reviewer #3: I Don't Know

3. Have the authors made all data underlying the findings in their manuscript fully available?

Reviewer #1: Yes

Reviewer #2: Yes

Reviewer #3: No

4. Is the manuscript presented in an intelligible fashion and written in standard English?

Reviewer #1: No

Reviewer #2: Yes

Reviewer #3: Yes

5. Review Comments to the Author

Reviewer #1: The authors present a novel project that endeavors to use machine leaning and multivariate to develop a better assessment of neuromotor recovery after limb transplant or nerve resection in a mouse model. There is good information presented, but enthusiasm for the manuscript is reduced due to a number of potentially correctable factors.

1. It isn't clear if this is a project focused on improving assessment of nerve injury patients and VCA recipients, or a project seeking to implement novel machine learning techniques. The manuscript as presented doesn't completely do either.

2. There are multiple grammar/usage errors. For example, in the abstract on line 74, "Not all interventions can be experimented in humans". I assume the authors mean not all interventions can be studied in humans. The following sentences which state "methods for evaluating rodent gait lack the multivariate description applied to humans limiting their effectiveness. The authors then go on to say that multivariate evaluation of murine gait models is a novel clinical problem. This statement is not clear, and examples like this are common in the submitted manuscript.

3. There is no clear hypothesis that will be tested. This should be added to the Objective of the abstract. Was the hypothesis that multi-variate analysis is better than univariate analysis? Why was a hind limb transplant model used? Did the authors hypothesize that multivariate analysis would detect deficits better than univariate analysis in transplants than in nerve resection models? Was that the case? The only proof I saw was that more factors were identified by a more complex analysis. Does that empirically mean it is better?

4. The authors need a better description in the methods section of what machine learning, and training of classifier models is and what it means in relation to normal animals, nerve transection and hind limb transplant models.

5. Based on clinical experience to date, it is unlikely that clinical lower limb transplantation will become common. Results with lower limb prosthetics are just too good. The authors might want to relate how the proposed work might benefit upper limb transplantation.

6. Pay attention to grammar, another example, line 138 - "peripheral nervous injury" - should be peripheral nerve injury

7. Be careful about rationale for using male mice in methods section. The statement that male mice were used, because in the author's experience "they appear to be more tolerant to hindlimb transplant surgery" is a difficult statement to defend.

8. Figure 2 - comparing normal function to nerve resection or a transplant and looking for differences is not difficult. The authors state that only 44% had a p value below the 0.001 threshold. Why is this not higher? State clearly what the authors expected and why it is important to show this data.

9. The same lack of purpose is exhibited in table 1. Is the point that there are more features with multivariate analysis? The statement "Within each type of of analysis (univariate vs. multivariate), the two pathological states have high degree of overlap with key differences among both pathological states" is very confusing. What are the authors trying to say?

10. Please go through the rest of the manuscript and clearly state the reason for the analysis, and how it relates to the hypothesis that will be presented in the abstract.

11. In figure 5, Why do healthy animals have scores close to no function, (1 = fully functional, 0 = no function)? Figure 5 also highlights the fact hat nerve function after only two weeks of healing is very limited. Better nerve function would be seen at one month. Did the authors want animals with incomplete nerve recovery, and if so, what was the rationale?

Reviewer #2: COMMENTS TO THE AUTHORS

1. The introduction of the article is quite long and needs to be shortened.

2. The hypothesis of the article should be stated more clearly in the introduction.

3. There seems to be an inconsistency between the title of the study and the hypothesis of the study. While the study title states "Multivariate description and scoring of neuromotor changes in a mouse model of peripheral nerve injury", the study seems to test the validity and reliability of a video analysis program called DigiGait. Therefore, it seems necessary to change either the title or the hypothesis of the article or the content of the study.

4. It is not understood why isotransplantation was preferred over autotransplantation in the study. However, autotransplantation is the first choice in limb amputations in the clinic.

5. In addition, histopathologic examination methods of neural recovery were not included in the study. Therefore, the study does not include detailed recovery data including the histopathological and biochemical examinations of the subjects who underwent transsection and limb transplantation.

6. Many other neuromotor analysis methods (such as swimming, sciatic function test, and climbing) were also not included in the study. In addition, target organ (such as gastrocnemius muscle) involvement levels were not included in the study. Therefore, detailed analysis of neuromotor recovery was not performed.

7. In addition, there are already studies similar to this study in the literature as follows:

a. Ganguly A, McEwen C, Troy EL, Colburn RW, Caggiano AO, Schallert TJ, Parry TJ. Recovery of sensorimotor function following sciatic nerve injury across multiple rat strains. J Neurosci Methods. 2017 Jan 1;275:25-32. doi: 10.1016/j.jneumeth.2016.10.018. Epub 2016 Oct 29. PMID: 27984099.

Reviewer #3: Thank you so much for providing me to review this manuscript. It seems that its aim is to use multivariate statistics to characterize and distinguish among etiologies of gait deficit in animals.

Below are some of the comments and suggestions for the manuscript:

• The first sentence in the abstract (lines 72-73) is not clear. “Quantitative scoring enabling comparison of neuromotor recovery in peripheral nerve injury models is critical to understanding and gauging the extent of injury and repair.” Just add commas to the sentence to be more readable and to separate the introductory phrase from the main clause. “Quantitative scoring, enabling comparison of neuromotor recovery in peripheral nerve injury models, is critical to understanding and gauging the extent of injury and repair.”

• Lines 97-98 are not enough for the conclusion. You may need to add briefly the contributions and advantages for your proposed solution.

• The introduction was written in a clear way, and the authors presented several works in the literature. It is better to add at least a recent reference from the last year.

• You should add references to the two approaches used in Lines 227-229.

• In line 254, you mentioned that the same animal was not included in both the training and test sets within a respective fold. How did you ensure that using randomly 10-fold cross validation?

• Authors should describe the dataset in more details. How many samples in the dataset? What is the number of samples of each class (healthy, surgery)? What is the number of mice that participated in the experiment? These are not mentioned in the methodology section.

• Authors should describe in more detail the data that is processed in the classifiers. How many input characteristics are there?

• What are the hyperparameters that were used in each classifier (e.g. C and gamma in SVM, number of trees in Random Forests…etc.)?

• Did you perform hyperparameter tuning to select the best parameters for each model? This may improve the performance.

• It is better to add the confusion matrix of the results and appropriate evaluation metrics.

• Maybe it is worth testing other ensemble-based classifiers (e.g. XgBoost or stacking classifiers) to improve the performance.

I hope that these comments are useful for you going forwards.

6. PLOS authors have the option to publish the peer review history of their article (what does this mean?). If published, this will include your full peer review and any attached files.

Reviewer #1: No

Reviewer #2: No

Reviewer #3: No

---

## [Author Response · Author response to Decision Letter 0]

16 May 2024

Please see attached "Response to Reviewers" file

---

## [Decision Letter · Decision Letter 1]

11 Sep 2024

PONE-D-23-36233R1Multivariate description of gait changes in a mouse model of peripheral nerve injury and traumaPLOS ONE

Dear Dr. Zhang,

Thank you for submitting your manuscript to PLOS ONE. After careful consideration, we feel that it has merit but does not fully meet PLOS ONE’s publication criteria as it currently stands. Therefore, we invite you to submit a revised version of the manuscript that addresses the minor points raised during the review process.

We look forward to receiving your revised manuscript.

Kind regards,

Annesha Sil, Ph.D.

Associate Editor

PLOS ONE

Journal Requirements:

Reviewers' comments:

Reviewer's Responses to Questions

**Comments to the Author**

1. If the authors have adequately addressed your comments raised in a previous round of review and you feel that this manuscript is now acceptable for publication, you may indicate that here to bypass the “Comments to the Author” section, enter your conflict of interest statement in the “Confidential to Editor” section, and submit your "Accept" recommendation.

Reviewer #4: All comments have been addressed

2. Is the manuscript technically sound, and do the data support the conclusions?

Reviewer #4: Yes

3. Has the statistical analysis been performed appropriately and rigorously? 

Reviewer #4: Yes

4. Have the authors made all data underlying the findings in their manuscript fully available?

Reviewer #4: Yes

5. Is the manuscript presented in an intelligible fashion and written in standard English?

Reviewer #4: Yes

6. Review Comments to the Author

Reviewer #4: Well written study addressing a notable gap in current PNI research and incorporation of machine learning to further minimize observational bias. The authors propose that a multivariate analysis of gait could uncover significant connections between spatiotemporal aspects of gait that are biologically pertinent to PNI. They further propose that these relationships will enable a more precise identification of distinct gait patterns compared to using only univariate analysis.

The authors have addressed all previous comments and concerns raised by the reviewers. Yet, concern still remains in generalizing the authors hypothesis to current gait research protocols within the scientific community. This issue may benefit from further discussion.

7. PLOS authors have the option to publish the peer review history of their article (what does this mean?). If published, this will include your full peer review and any attached files.

Reviewer #4: No

---

## [Author Response · Author response to Decision Letter 1]

13 Sep 2024

Dear Editor,

Thank you for the reviewers’ comments and suggestions. Our responses are the following:

Review Comments to the Author

Reviewer #4: Well written study addressing a notable gap in current PNI research and incorporation of machine learning to further minimize observational bias. The authors propose that a multivariate analysis of gait could uncover significant connections between spatiotemporal aspects of gait that are biologically pertinent to PNI. They further propose that these relationships will enable a more precise identification of distinct gait patterns compared to using only univariate analysis.

The authors have addressed all previous comments and concerns raised by the reviewers. Yet, concern still remains in generalizing the authors hypothesis to current gait research protocols within the scientific community. This issue may benefit from further discussion.

Response: 

We are grateful for the opportunity to further discuss the generalizability of the hypothesis to current gait research protocols within the scientific community. We have added a few paragraphs to the Discussion section to address this as excerpted below. The additions also include references to a number of studies that help build support for the potential generalizability of the approach:

(page 24) “The level of exploratory statistical description in the prior few paragraphs is largely absent from rodent studies across the field, including some of the latest that employ gait analysis [18, 25, 37]. Thus, we hypothesize that multivariate statistics has generalizable potential in reducing observational bias to current pre-clinical gait research protocols in the scientific community.

Towards this hypothesis, a recent study contributes evidence toward using gait as a quantitative translational outcome metric for therapeutic development in Angelman syndrome and other genetic neurodevelopmental syndromes [25]. This study contributes a comprehensive, multivariate characterization pipeline for application in the study of any pathologies in which gait is a quantitative translational outcome metric. Another study was comparable in its level of statistical description [31], coincidentally doing so in a genetic neurodevelopmental model of cerebellar ataxia [31]. Lambert et al. supported the value of multivariate characterization of gait by conducting feature extraction on highly dimensional gait data from a central nerve injury model of cerebellar ataxia. The authors hypothesized certain indirect, common factors that characterize gait and underlie the more directly measurable features. These common factors included thrust, rhythmicity, and contact area and were directly useful in discriminating between animals with a central lesion vs. those without [31]. In this study, we observed potential patterns in the selected features and hypothesized that pathological gait due to traumatic nerve injury could also be characterized by certain latent factors that are determined by and composed of the selected features.

(page 26) “This study offers a methodology that may be applied to existing DigiGait datasets to conduct similar exploratory multivariate analysis to reduce observational bias from traditional univariate methods. Evaluating the generalizability of this methodology on other etiologies of gait deficit is the subject of a subsequent study… As a new methodology within rodent gait research protocols, one limitation of this study is its limited scope. On one hand, the multivariate statistics demonstrated in this study offer a convenient way for researchers to conceptualize meaningful groupings of features. On the other hand, clear biological tie-ins and strong statistical significance in classifier performance when applied to gait research protocols across the scientific community stands to be evaluated. We have identified datasets of DigiGait data available at our Behavioral Phenotyping Core from investigators studying central causes of gait disorder (e.g. stroke, transient ischemic attack) for us to apply the pipeline to and are pursuing said inquiry…”

---

## [Decision Letter · Decision Letter 2]

7 Oct 2024

Multivariate description of gait changes in a mouse model of peripheral nerve injury and trauma

PONE-D-23-36233R2

Dear Dr. Zhang,

We’re pleased to inform you that your manuscript has been judged scientifically suitable for publication and will be formally accepted for publication once it meets all outstanding technical requirements.

Kind regards,

Antal Nógrádi, M.D., Ph.D., D.Sc.

Academic Editor

PLOS ONE

Additional Editor Comments (optional):

Reviewers' comments:

Reviewer's Responses to Questions

**Comments to the Author**

1. If the authors have adequately addressed your comments raised in a previous round of review and you feel that this manuscript is now acceptable for publication, you may indicate that here to bypass the “Comments to the Author” section, enter your conflict of interest statement in the “Confidential to Editor” section, and submit your "Accept" recommendation.

Reviewer #4: All comments have been addressed

2. Is the manuscript technically sound, and do the data support the conclusions?

Reviewer #4: Yes

3. Has the statistical analysis been performed appropriately and rigorously? 

Reviewer #4: Yes

4. Have the authors made all data underlying the findings in their manuscript fully available?

Reviewer #4: Yes

5. Is the manuscript presented in an intelligible fashion and written in standard English?

Reviewer #4: Yes

6. Review Comments to the Author

Reviewer #4: All of the reviewers concerns have been addressed by the authors. I have no further unaddressed comments

7. PLOS authors have the option to publish the peer review history of their article (what does this mean?). If published, this will include your full peer review and any attached files.

Reviewer #4: No

---

## [Editor Report · Acceptance letter]

4 Nov 2024

PONE-D-23-36233R2 

PLOS ONE

Dear Dr. Zhang, 

I'm pleased to inform you that your manuscript has been deemed suitable for publication in PLOS ONE. Congratulations! Your manuscript is now being handed over to our production team.

Kind regards, 

on behalf of

Prof. Antal Nógrádi 

Academic Editor

PLOS ONE